# The Potential Influence of Uremic Toxins on the Homeostasis of Bones and Muscles in Chronic Kidney Disease

**DOI:** 10.3390/biomedicines11072076

**Published:** 2023-07-24

**Authors:** Kuo-Chin Hung, Wei-Cheng Yao, Yi-Lien Liu, Hung-Jen Yang, Min-Tser Liao, Keong Chong, Ching-Hsiu Peng, Kuo-Cheng Lu

**Affiliations:** 1Division of Nephrology, Department of Medicine, Min-Sheng General Hospital, Taoyuan City 330, Taiwan; corey926@gmail.com; 2Department of Pharmacy, Tajen University, Pingtung 907, Taiwan; 3Department of Anesthesiology, Min-Sheng General Hospital, Taoyuan City 330, Taiwan; M000924@e-ms.com.tw; 4Department of Medical Education and Clinical Research, Min-Sheng General Hospital, Taoyuan City 330, Taiwan; 5Department of Family Medicine, Min-Sheng General Hospital, Taoyuan City 330, Taiwan; M002835@e-ms.com.tw; 6Department of General Medicine, Min-Sheng General Hospital, Taoyuan City 330, Taiwan; prestontan@share-hope.com; 7Department of Pediatrics, Taoyuan Armed Forces General Hospital Hsinchu Branch, Hsinchu City 300, Taiwan; liaoped804h@yahoo.com.tw; 8Department of Pediatrics, Taoyuan Armed Forces General Hospital, Taoyuan 325, Taiwan; 9Department of Pediatrics, Tri-Service General Hospital, National Defense Medical Center, Taipei 114, Taiwan; 10Division of Endocrinology and Metabolism, Department of Medicine, Min-Sheng General Hospital, Taoyuan City 330, Taiwan; m001119@e-ms.com.tw; 11Division of Nephrology, Taipei Tzu Chi Hospital, Buddhist Tzu Chi Medical Foundation, and School of Medicine, Tzu Chi University, Hualien 970, Taiwan; kuochenglu@gmail.com; 12Division of Nephrology, Department of Medicine, Fu-Jen Catholic University Hospital, School of Medicine, Fu-Jen Catholic University, New Taipei City 242, Taiwan; 13Division of Nephrology, Department of Medicine, Tri-Service General Hospital, National Defense Medical Center, Taipei 114, Taiwan

**Keywords:** uremic toxins, bone loss, sarcopenia, chronic kidney disease, indoxyl sulfate

## Abstract

Patients with chronic kidney disease (CKD) often experience a high accumulation of protein-bound uremic toxins (PBUTs), specifically indoxyl sulfate (IS) and p-cresyl sulfate (pCS). In the early stages of CKD, the buildup of PBUTs inhibits bone and muscle function. As CKD progresses, elevated PBUT levels further hinder bone turnover and exacerbate muscle wasting. In the late stage of CKD, hyperparathyroidism worsens PBUT-induced muscle damage but can improve low bone turnover. PBUTs play a significant role in reducing both the quantity and quality of bone by affecting osteoblast and osteoclast lineage. IS, in particular, interferes with osteoblastogenesis by activating aryl hydrocarbon receptor (AhR) signaling, which reduces the expression of Runx2 and impedes osteoblast differentiation. High PBUT levels can also reduce calcitriol production, increase the expression of Wnt antagonists (SOST, DKK1), and decrease klotho expression, all of which contribute to low bone turnover disorders. Furthermore, PBUT accumulation leads to continuous muscle protein breakdown through the excessive production of reactive oxygen species (ROS) and inflammatory cytokines. Interactions between muscles and bones, mediated by various factors released from individual tissues, play a crucial role in the mutual modulation of bone and muscle in CKD. Exercise and nutritional therapy have the potential to yield favorable outcomes. Understanding the underlying mechanisms of bone and muscle loss in CKD can aid in developing new therapies for musculoskeletal diseases, particularly those related to bone loss and muscle wasting.

## 1. Introduction

In chronic kidney disease (CKD), the kidneys gradually lose their ability to excrete toxic waste into the urine, leading to a buildup of serum uremic toxins in the body [1]. CKD patients may experience higher fracture and mortality rates, associated with CKD–mineral bone disease (CKD–MBD), characterized by abnormal bone turnover and reduced mineralization and volume [2]. In patients with early CKD (stages 2–3), wingless (Wnt) signaling inhibitors, such as Dickkopf-1 (DKK1), sclerostin (SOST), and secreted frizzled-related protein (sFRP), are secreted from the kidney or osteocytes in bone or calcified soft tissue. These inhibitors can affect osteoblasts, leading to a decrease in their viability. Additionally, the retention of protein-bound uremic toxins (PBUTs) such as indoxyl sulfate (IS)/p-cresyl sulfate (pCS) further impairs the viability and function of both osteoblasts and osteoclasts [3]. CKD leads to alterations in the gut microbiome (dysbiosis) and the accumulation of gut-derived uremic toxins in plasma [4]. It has previously been shown that the uremic toxins IS and pCS are inversely correlated to the estimated glomerular filtration rate (eGFR) across all stages of CKD [5,6]. IS is associated with low bone turnover. It suppresses bone formation by downregulating the parathyroid hormone receptor (PTHR) [7], inhibits Wnt signaling [8], and promotes apoptosis [9] in cultured osteoblast cells. The administration of an intestinal adsorbent, which reduces uremic toxins, increases bone turnover in uremic rats [10]. The term “uremic osteoporosis” is used to describe the loss of bone quality with normal bone mass due to the effects of PBUTs on bone health [3]. When renal function progressively deteriorates, metabolic acidosis and hyponatremia contribute to bone loss [11]. In patients with CKD stages 4–5, the dysregulation of calcium, phosphate, vitamin D, and parathyroid hormone (PTH) can lead to varying levels of PTH [12]. High-turnover bone disease develops (CKD stages 4–5) when high serum PTH levels overcome peripheral PTH resistance and other inhibitory factors of bone formation [13]. High levels of PTH in progressive CKD upregulate the receptor activator of nuclear factor-kappa B ligand (RANKL) mRNA and inhibit osteoprotegerin (OPG) gene expression in bone marrow stromal osteoblasts [14], leading to increased quantities of osteoclasts and osteoblasts [15]. High PTH levels drive the indolent OB into high viability and function, but poor quality in behavior, resulting in both bone quality and quantity loss [13]. However, with the medical or surgical treatment of secondary hyperparathyroidism (SHPT), which involves the removal of the stimulator of PTH, the bone cells may return to the innate low bone cell viability status, the low bone turnover disorders [12] (Figure 1).

Redox signaling alterations contribute to cachexia in CKD due to factors such as uremic toxins, inflammation, and metabolic/hormonal changes, inducing oxidative stress, which leads to bone loss and muscle wasting [16,17]. A mild accumulation of PBUTs significantly contributes to muscle wasting in early-stage CKD (stages 2–3). Exercise may help counteract these effects [16]. Antioxidant therapies have shown positive effects in experimental models of CKD. However, treating CKD-associated bone loss and muscle wasting remains challenging [16,18]. The molecular mechanisms involved include an imbalance between protein degradation and synthesis, increased levels of reactive oxygen species (ROS) and inflammatory cytokines, activation of myostatin and atrogene expression, impaired mitochondrial function, and the negative effects of PBUTs, parathyroid hormone, and angiotensin II on muscle mass and endurance [19,20]. IS is implicated in stimulating ROS-mediated myostatin and atrogene expression and impairing mitochondrial function [21].

In advanced stages of CKD (stages 4–5), the occurrence of cachexia becomes prominent. Cachexia is characterized by an increase in energy expenditure and the loss of both muscle and adipose tissues [22]. In addition to increasing levels of PBUTs, excessive levels of PTH have harmful effects on skeletal muscle metabolism, leading to worsened muscle weakness and atrophy [23]. In primary hyperparathyroidism, Pattern et al. found muscle atrophy in both type I and type II muscle fibers, with a more pronounced effect on type II fibers [24]. Profound changes in muscle gene expression have been observed, potentially contributing to clinical manifestations, such as muscular fatigue [25,26]. Clinical study findings indicate that high PTH levels increase the risk of sarcopenia in elderly women, and individuals with hyperparathyroidism have a higher prevalence of sarcopenia compared with those without [27]. Age-related elevation in PTH levels may contribute to muscle strength and mass loss in sarcopenia [28]. PTH and PTHrP (PTH-related peptide) induce white adipose tissue browning; activate thermogenic and atrophy-related genes, such as uncoupling protein 1 (UCP-1), atrogin-1, muscle RING-finger protein-1 (MuRF1); and upregulate the ubiquitin–proteasome proteolytic system (UPS), leading to muscle wasting [29,30]. In CKD and cancer, PTH and tumor-derived PTHrP drive adipose tissue browning and cachexia [31]. Targeting the PTH/PTHrP pathway holds therapeutic potential for treating cachexia [32].

Based on the varying effects of PBUTs and PTH levels on bone turnover and muscle wasting in CKD, we conducted this review to explore the specific impact of PBUTs on the homeostasis of bones and muscles.

## 2. Pathophysiology of PBUTs: IS and pCS

PBUTs accumulate in CKD patients due to impaired renal clearance. These solutes, which are normally excreted by healthy kidneys, accumulate in CKD due to reduced filtration, impaired tubular secretion, harmful gut microbial metabolism, and unfavorable dietary precursor [33]. PBUTs, such as IS and pCS, tightly bind to proteins such as serum albumin, making their removal from the body challenging [34].

### 2.1. Metabolism of PBUTs

Indole is a byproduct of tryptophan metabolism by gut bacteria. After absorption from the intestines, indole is transported to the liver, where it is hydroxylated and sulfurated, becoming IS, which enters the bloodstream and is taken up by renal tubular cells via organic anion transport 1 (OAT1) and 3 transporters [35]. Elevated IS levels can amplify oxidative stress, free radicals, and inflammation [36].

Our previous research proposed that IS acts on the aryl hydrocarbon receptor (AhR), increasing the production of arachidonic acid 12-lipoxygenase and endovanillin 12(S)-hydroxyeicosatetraenoic acid, leading to the enhanced functionality of transient receptor potential vanilloid 1 (TRPV1) and inducing renal tubular injury [37]. Studies using AhR knockout and α-naphthoflavone inhibitors have also shown that AhR signaling plays a role in diabetic nephropathy, causing oxidative stress injury, mesangial cell activation, extracellular matrix accumulation, and macrophage infiltration due to AhR nuclear translocation and increased toxic bioproducts [38]. Uremic toxin/AhR signaling harms the cardiovascular system in CKD patients.

The protein-bound uremic toxin pCS is produced by gut bacteria that metabolize tyrosine and phenylalanine and is usually eliminated through the kidneys [39]. pCS induces nicotinamide adenine dinucleotide phosphate (NADPH) oxidase to increase ROS production, which leads to the production of inflammatory cytokines in tubular epithelial cells, causing progressive renal fibrosis [40]. High serum pCS concentrations have been associated with higher hospitalization rates and adverse clinical outcomes [41], aggravated clinical symptoms of uremia [42], and cardiovascular disease in numerous studies [43,44].

### 2.2. Pathogenesis of PBUTs

#### 2.2.1. AhR Signaling

Regulating the immune system, metabolic diseases, and bone health, the AhR receptor demonstrates its responsiveness to endogenous toxins or environmental pollutants [45]. Upon agonist binding, such as IS, AhR dissociates from its chaperone complex and translocates to the nucleus. It then forms a complex with the aryl hydrocarbon receptor nuclear translocator (ARNT) and activates the expression of target genes, including CYP1A1 and CYP1B1, by binding to the xenobiotic response element (XRE) sequence in the canonical pathway [46]. Unliganded AhR acts as a proteasome degradation enzyme with ligand-dependent E3 ubiquitin ligase activity and targets protein ubiquitination, such as nuclear factor of activated T cells 1 (NFATc1), the main transcription factor for osteoclast differentiation [47]. In the noncanonical pathway, AhR interacts with Kruppel-like transcription factor 6 (KLF6) and activates the expression of target genes, such as plasminogen activator inhibitor-1 (PAI-1) and p21^cip1^, by binding to a nonconsensus XRE nucleotide sequence [48,49].

IS and kynurenine (KYN), PBUTs produced by gut microbiota, can increase oxidative stress, increase inflammation, and damage cells in the body [50,51,52]. As CKD advances, increased exposure to uremic toxins can lead to the overexpression of tissue inhibitors of osteopontin, transforming growth factor beta 1(TGF-β1), metalloproteinase 1, and endothelin 1. This, in turn, induces epithelial–mesenchymal transition, causing damage to both the cardiovascular system and bones [53]. Research in toxicology has shown that AhR ligands can inhibit the differentiation of osteoblasts [45,54]. Nguyen et al. showed that the activation of AhR by tetrachlorodibenzo-p-dioxin (TCDD) inhibits the osteoblastic differentiation of bone marrow–derived stem cells [55]. In a previous study, it was observed that exposure to TCDD in wild-type mice led to impaired mechanical bone formation and increased trabecular bone volume. Modifying the structure of the AhR transcriptional activation domain by TCDD leads to the disruption of bone remodeling, which ultimately reduces bone strength [56]. TCDD modifies the structure of the AhR transcriptional activation domain, disrupting bone remodeling and ultimately reducing bone strength [57]. Conversely, resveratrol, an AhR antagonist, increases bone mineral density (BMD) and bone volume [58]. TCDD and benzo(a)pyrene (BaP) smoke toxins activate AhR to accelerate osteoclast bone resorption by activating cytochrome P450 1a/1b enzymes [59]. Nevertheless, due to the utilization of a substantial amount of smoke toxins in this study, the results obtained may not be directly transferable to a clinical environment. On the contrary, Voronov et al. demonstrated that BaP exerts an inhibitory effect on osteoclast activity by influencing RANKL concentration, osteoclast precursors, and stromal cell density [60]. A higher level of RANKL may overcome the inhibitory effect of BaP on AhR signaling activation.

In 1976, a chemical factory explosion in Seveso, Italy, exposed residents to high levels of TCDD [61]. A study involving 48 children who were under 9.5 years old at the time found that elevated TCDD levels were associated with tooth development issues, such as enamel problems and missing incisors [62]. However, Eskenazi et al. did not find any significant associations between TCDD levels and measures of bone mass density, including the spine, total hip, and femoral neck [63]. Another study also did not find any significant links between TCDD levels and bone density measures. The long-term effects and accumulation of TCDD in bones remain uncertain, primarily because of its estimated half-life in the body [64]. In this manuscript, we examine the impact of TCDD on exogenous AhR signaling and its potential consequences for CKD patients. Additional research is necessary to comprehensively grasp the clinical importance of extended TCDD exposure in such scenarios.

In a mouse model of collagen-induced arthritis, KYN binds to AhR, activating the extracellular regulated protein kinase (ERK) signaling pathway and inhibiting osteoblast activity [58,65]. KYN-treated osteoclasts promote the expression of CYP1A1, intensifying bone resorption and enhancing osteoclast activity [66,67]. These studies suggest that AhR ligands have varying effects, depending on their concentration and duration of exposure.

#### 2.2.2. PBUTs Enhance ROS

IS activates NADPH oxidase, leading to increased oxidative stress via ROS production [68]. High serum IS promotes oxidative stress and inflammatory gene expression in renal tubular cells by inducing free radical production [36].

IS stimulates chemokine and transforming growth factor-1 (TGF-1) production by renal tubular cells, leading to increased free radical production, oxidative stress, cytokine production, inflammation [36], and progression of CKD [69]. Increased free radicals reinforce the activity of the tissue inhibitor of metalloproteinase-1 (TIMP-1) and nuclear factor kappa B. This, in turn, increases the potency of PAI-1, which is known to exacerbate interstitial fibrosis and tubular atrophy [69,70].

In CKD patients, IS activates monocyte-mediated inflammation and adipocyte secretion of tumor necrosis factor-α (TNF-α) and interleukin (IL)-6 through oxidative stress [71]. A previous article showed that higher levels of IS and indole-3-acetic acid (IAA) in CKD patients may be related to the development of cardiovascular disease. These factors induce higher levels of IL-6 and monocyte chemotactic protein-1 (MCP-1) expression in monocytes [72]. Kim et al. showed that IS activates oxidative stress and acts as an osteotoxin in primary osteoblast cultures [9]. Free radicals disrupt the redox status of mesenchymal cells and activate mitogen-activated kinase proteins, leading to mesangial cell proliferation [73]. In vivo, IS administration decreases the restorative activity of superoxides in normal and CKD rats [74], ultimately causing glomerulosclerosis, renal interstitial fibrosis, and accelerated deterioration of renal function [69].

#### 2.2.3. PBUTs Diminish the Synthesis of Nitric Oxide (NO)

IS has multiple effects on vascular tissues and endothelial cells. In rat aorta, it reduces NO/cGMP signaling, leading to increased contraction in response to endothelin-1 [75]. In the human umbilical cord, IS modifies endothelial cell tight junctions by reducing endothelial cadherin and diminishing endothelial NO synthase activity [76]. Additionally, previous studies suggest that IS enhances O_2_^−^ production, induces the generation of ROS, and inactivates NO in vascular tissue through the AhR–NADPH oxidase pathway, resulting in weakened vasodilation [77]. These effects are associated with increased oxidative stress and cellular senescence, which lead to decreased NO production and endothelial dysfunction [78].

#### 2.2.4. The Epigenetic Effects of PBUTs

PBUTs have been studied for their phenotypic and molecular mechanisms, including epigenetic effects. They can inhibit klotho gene expression, which may affect kidney function [79]. PBUTs inhibit klotho gene expression by increasing DNA methyltransferase (DNMT) expression and DNA methylation. PBUTs also contribute to the expression of DNMT-1, 3a, and 3b [80], which regulate telomere expression and telomerase activity. DNMT-1 is the most important enzyme for DNA methylation. Uremic toxin exposure increases DNMT protease expression, leading to hypermethylation and alternative splicing of the klotho gene, impairing normal physiological function as mammalian DNMT-1 is essential for DNA methylation [80,81]. The use of DNMT inhibitors can reduce klotho gene methylation, leading to improved klotho mRNA and protein expression.

Klotho acts as a coreceptor for FGF 23 (fibroblast growth factor 23) and is primarily present in the cerebral choroid plexus and renal tubules [82]. Although klotho’s protective effect on renal function is documented, its physiological function and mechanism remain unclear [83]. Klotho contributes to the suppression of the renal reabsorption of phosphate and vitamin D biosynthesis [82]. It also plays a role in the antiaging process [84] and acts as a sensitive biomarker of kidney disease, while also being a protective factor in kidney disease deterioration, disturbed mineral metabolism, and vascular calcification of CKD [85].

Furthermore, klotho inhibits the insulin-like growth factor (IGF-1) and activates TGF-1 signaling pathways, which leads to a reduction in renal fibrosis [86]. Skeletal muscle IGF-1 regulates protein synthesis and catabolic pathways by increasing protein synthesis through signaling pathways PI3K (phosphoinositide 3-kinase)/Akt (protein kinase B)/mTOR (mammalian target of rapamycin) and PI3K/Akt/GSK3β (glycogen synthase kinase-3 beta). Activating PI3K/Akt can prevent the increase in FoxO (forkhead box protein O) and inhibit the transcription of the E3 ubiquitin ligase, which regulates the ubiquitin–proteasome system (UPS), thus avoiding proteolysis [87]. IS and pCS can increase DNMT activity via the Ras–MEK (ERK kinase) pathway, methylate the klotho gene, and accelerate the progression of renal function [80]. Targeting specific genes could be a potential novel therapy for halting the progression of abnormal bone homeostasis in CKD.

## 3. Effects of PBUTs on Bone

### 3.1. PBUTs Influence Bone Metabolism

Exposure to PBUTs associated with CKD and a sedentary lifestyle can impact bone metabolism. These factors can contribute to the conversion of white fat cells into brown fat cells, the recruitment of immune cells, and the release of proinflammatory cytokines, leading to chronic inflammation [34]. However, engaging in physical activity can counteract these effects by inducing sarcomeric contracture in muscles, which releases anti-inflammatory muscle factors [88]. Various myokines, such as IL6, irisin, IGF-1, BDNF (brain-derived neurotrophic factor), FGF2, and myostatin, can have either anabolic or catabolic effects on bones. Additionally, osteocalcin enhances muscle anabolism, while sclerostin attenuates muscle catabolism [89]. The browning of adipocytes stimulates thermogenesis through the release of lipolytic myokines, and the activation of adipokines, such as TNF-α, adiponectin, resistin, and leptin, can modulate muscle and bone metabolism [90]. The interactions between PBUTs, physical activity, myokines, and adipokines influence the complex relationship between bone health and overall metabolism.

#### 3.1.1. Uremic Toxin Exposure Affects Osteoclastogenesis

Recent research revealed that AhR acts as a transcription factor for the OC precursor, and low serum IS levels enhance the expression of NFATc1 (nuclear factor of activated T cells 1) [91]. Low-dose IS boosts both the transcriptional activity of AhR and the production of NFATc1, while reducing NFATc1 ubiquitination by decreasing AhR E3 ligase activity. Conversely, higher IS levels stimulate the ligase pathway of AhR E3 ubiquitin, decrease NFATc1 expression, and increase NFATc1 ubiquitination [92]. IS concentrations suppress ARNT expression, enhance NFATc1 ubiquitination, and inhibit osteoclast differentiation and function through the Akt, JNK (Jun NH2-terminal kinase), ERK1/2, and p38 pathways [93].

AhR plays a significant role in bone remodeling by regulating osteoclast and osteoblast interactions [94]. AhR knockout in mice increases bone mass and reduces bone resorption [58]. Low levels of active AhR were found to increase bone resorption in female mice in previous studies [95]. Additionally, IS inhibits RANKL production by osteoblasts, which leads to the suppression of RANKL-dependent preosteoclast differentiation into mature osteoclasts [96].

ARNT, a hypoxia-inducible transcription factor, protects renal function by inducing the transcription of ALK3 (bone morphogenetic protein receptor type 1A), which is the principal mediator for antifibrotic and pro-regenerative responses of BMP signaling [97]. However, high levels or prolonged exposure to IS suppresses ARNT expression, disrupting normal bone metabolism. AhR antagonists may be used as novel medications to treat renal osteodystrophy and muscle wasting in CKD patients by preventing this interference.

#### 3.1.2. PBUTs Impair Osteoblastogenesis

PBUTs, including IS, are absorbed by osteoblasts through the OAT-3 membrane, leading to the generation of free radicals and dysfunction of osteoblasts [7]. This dysfunction includes reduced PTHR expression, which may cause PTH resistance in bones, ultimately resulting in low bone turnover in dialysis patients [7].

Runx2 is a major transcription factor for osteoblast differentiation and regulates bone formation [98]. Mechanical stress activates the MAPK (mitogen-activated protein kinase), Ras/Raf-dependent ERK1/2, and p38 MAPK signaling pathways, which favor osteoblast differentiation through the activation of Runx2, as shown by Kanno et al. [99]. AhR ligands inhibit osteoblastic differentiation, as observed by Korkalainen et al. [45]. The reduction of osteoblastic differentiation is caused by IS via the p38 MAPK/Runx2 and AhR/ERK1/2 pathways.

#### 3.1.3. PBUTs Reduce Bone Mass

IS reduces bone cell activity and promotes apoptosis in osteoblasts [9]. IS inhibits the differentiation of osteoblasts, as indicated by decreased levels of osteonectin, type 1 collagen, and alkaline phosphatase in mouse calvaria preosteoblastic cells (MC3T3-E1) as osteoblast cell line culture [8]. Moreover, IS not only inhibits the function of osteoblasts but also inhibits the maturation of osteoclasts, which could affect bone remodeling in CKD patients [93]. Experiments on bone mass have demonstrated that even a short exposure period (2 months) can lead to reduced bone mass, which is related to the accumulation of uremic toxins rather than bone metabolism or parathyroid function [100].

IS decreases the number of PTH receptors in osteoblasts and interferes with the generation of osteoblasts stimulated by cyclic adenosine monophosphate (cAMP), which leads to decelerated effects of PTH on the skeletal system [101]. This bone resistance to PTH causes not only high bone turnover in patients with high PTH levels but also low bone turnover in patients with low PTH levels. This may predispose the soft tissues and vascular system to calcification or ossification under hyperphosphatemia, hypercalcemia, and low vitamin D levels, accompanied by abnormal serum PTH levels [102]. Studies suggest that PBUTs have the potential to cause significant uremic osteoporosis, and even increase the fracture risk factor in people with CKD [3,103].

CKD patients with SHPT may experience either high or low bone turnover, leading to various bone-related problems. The two conditions require different treatments depending on the actual bone turnover status [12].

#### 3.1.4. PBUTs Reduce the Bone Quality

Patients with CKD often suffer from osteoporosis, which is characterized by reduced bone strength and bone mineral density with a T-score of 2.5 or lower according to the WHO [104]. Patients with CKD can develop uremic osteoporosis, which is characterized by low bone strength despite adequate bone mass due to high levels of PBUTs [3]. This is because bone strength depends not only on bone mass but also on bone quality, which is often overlooked [105].

IS and pCS contribute to the destruction of the protein and mineral structure of bone in CKD patients, resulting in bone mass imbalance. In a murine study, IS was found to alter the mechanical properties of broken bones, resulting in lower bone elasticity and an unbalanced ratio of minerals and phosphates [103]. IS concentrations were positively correlated with bone formation rate, osteoblast region, bone fibrosis volume, and osteoid volume in one study [106]. In experimental CKD animals, residual kidney function was inversely related to the mechanical elastic properties of bones, which were significantly associated with changes in bone biochemical content. PBUT can deteriorate bone material properties and disrupt bone elasticity [3]. Changes in bone quality mainly cause bone injury in patients with early CKD, often accompanied by low bone turnover.

#### 3.1.5. PBUTs Induce Bone Resistance to PTH

CKD patients frequently experience SHPT. The K/DOQI (National Kidney Foundation Kidney Disease Outcomes Quality Initiative) clinical practice guidelines recommend intact PTH serum concentrations of 150–300 pg/mL for end-stage renal disease (ESRD) and dialysis patients [107]. PBUTs, including IS, impair osteoblast function by inducing free radical production and reducing PTHR expression, resulting in decreased PTH response [7,108]. This can be attributed to the decrease in cAMP levels in the post-PTH receptor pathway and the accumulation of abnormal PTH fragments [8].

#### 3.1.6. PBUTs Disturb the Synthesis of Vitamin D

The reduced renal function increases FGF23, which lowers the activity of renal 1-hydroxylase and decreases the production of active vitamin D (1,25-(OH)_2_D_3_). Meanwhile, IS increases 24-hydroxylase activity, which degrades 1,25-(OH)_2_D_3_ and 25-hydroxyvitamin D, leading to insufficient serum concentrations of 1,25-(OH)_2_D_3_ and 25-(OH)D_3_ [109]. High uremic toxins and low active vitamin D in ESRD patients can reduce the nonrenal clearance of CYP3A4 substrates, including atorvastatin and erythromycin [110].

#### 3.1.7. PBUTs Affect the Differentiation of T Cells

PBUTs have toxic effects on T cells, contributing to chronic inflammation in CKD patients. IS affects various biological functions of T cells, including cell cycle regulation and inflammatory responses. IS-stimulated T cells upregulate AhR downstream target genes, such as NQO1 (NAD(P)H quinone dehydrogenase 1), AhR receptor, CYP1A1, and CYP1B1. These stimulated T cells also enhance the expression of the proinflammatory genes IFN-γ and TNF-α [111]. In the cytoplasm, IS enters T cells and triggers the activation of AhR [112], promoting the differentiation of T cells into T-helper 17 (Th17) cells, which play a role in adaptive immunity and inflammation. Th17 differentiation is induced by TGF-β, IL-1β, and IL-6 [113], while AhR contributes to Th17 cell differentiation by controlling Stat1 activation [114]. CKD patients experience oxidative stress, cardiovascular disease, and endothelial cell inflammation due to the abnormal activation of the immune system cascade through IS-activated AhR signaling [112,115].

Recent research demonstrated that bone-derived bone morphogenetic proteins (BMPs) have an impact on Treg cells (regulatory T cells). The activation of effector and Foxp_3_^+^ regulatory CD_4_^+^ T cells (Treg cells) upregulates BMP receptor 1α (BMPR1α), which in turn influences the functions of Th and Treg cells. The findings indicate that BMPR1α plays a role in inhibiting the generation of proinflammatory Th17 cells while supporting the maintenance of peripheral Treg cells [116].

## 4. IS and pCS on the Muscle

In catabolic diseases such as CKD, the presence of uremic toxins affects the muscle. In CKD, elevated levels of myostatin, TGF-β, and glucocorticoids contribute to increased expression of autophagy-related genes, atrogin-1, and muscle RING-finger 1 (MuRF1). As a result, this leads to a reduced activation of the insulin/IGF-1-Akt-mTOR pathway, resulting in diminished protein synthesis and muscle mass [117].

IS, which enters muscle cells through OATs, plays a role in stimulating the AhR and NADPH oxidase pathways [118]. This stimulation leads to the production of ROS. The excessive ROS production, combined with the overexpression of myostatin and atrogin-1, by triggering the production of inflammatory cytokines, IS contributes to muscle atrophy. Furthermore, ROS production negatively affects mitochondrial function, further impacting muscle health [119,120]. pCS, on the other hand, induces insulin resistance in muscles by reducing the capacity of insulin or IGF-1-IRS-AKT signaling through increased activity of ERK1/2. This interference with insulin signaling can impair protein synthesis and muscle growth [121,122]. Understanding these effects is crucial for developing targeted interventions to preserve muscle mass and function in patients with catabolic conditions (Figure 2).

### 4.1. Sarcopenia in CKD

Sarcopenia, characterized by muscle loss, is influenced by multiple risk factors. Pathophysiological mechanisms involving ubiquitin, insulin/IGF-1, myostatin, and IS contribute to muscle degradation and impaired regeneration [123]. In individuals with CKD, there is a correlation between declining renal function and reduced physical activity. Nutritional interventions and physical training have been shown to be beneficial in managing exercise intolerance associated with CKD [124]. In advanced stages of nephropathy, this condition is referred to as uremic sarcopenia [125]. Accelerated muscle loss in CKD patients is associated with lower quality of life, depression, malnutrition, cardiometabolic complications, and heightened risks of hospitalization and mortality [126,127]. The underlying causes for these effects primarily include decreased muscle strength and protein synthesis, as well as an increase in the breakdown of muscle proteins [128,129]. Various factors, including ubiquitin–proteasome system (UPS) activation, mediate myocyte apoptosis. These factors, such as angiotensin II, inflammation, neural/hormonal factors, and metabolic acidosis, impair insulin/IGF-I signaling. This abnormal signaling leads to muscle protein breakdown via UPS and caspase-3. Caspase-3 damages muscle structure, serving as a substrate for UPS. Activated caspase-3 boosts proteolysis by stimulating muscle protease activity [130]. Muscle cells from CKD patients exhibit heightened protein degradation and elevated cachexia markers (fbox32 and TRIM63). Moreover, they display anabolic resistance to IGF-1, hindering protein synthesis [131,132,133]. Research conducted in CKD patients [134,135] indicates a positive association between increasing levels of advanced glycation end products (AGEs) and oxidative stress, which negatively impacts muscle function. The presence of AGEs in CKD contributes to the development and progression of sarcopenia [136]. Although therapeutic drugs targeting muscle wasting mechanisms have been explored, most trials have focused on aged patients without CKD, and no drugs have been approved specifically for sarcopenia treatment yet [137].

### 4.2. PTH and Muscle Atrophy

PTH receptors, traditionally associated with bone and kidney, have been discovered to be highly expressed in nontraditional targets, such as the heart, liver, brain, and pancreas. Through its interaction with these receptors, PTH can impact cell function in these tissues, resulting in cAMP accumulation [30,138]. Kimura et al. conducted a study using a mouse cellular model and demonstrated that the physiological level of PTH plays a role in skeletal muscle regeneration. They emphasized the significance of PTHR1 expression and the use of PTH (1–34) in promoting myocyte differentiation, accelerating myogenesis, and facilitating myotube production [30,139]. However, when rats were treated with PTH for 4 days, their skeletal and cardiac muscles showed reduced mitochondrial oxygen consumption, increased ROS levels, and lower energy production. These unexpected findings contradicted the anticipated increase in energy production associated with PTH’s osteoanabolic effect on osteoblasts. These changes were attributed to enhanced entry and accumulation of calcium [140]. Patients with long-standing secondary and tertiary hyperparathyroidism may experience muscle dysfunction and wasting. The causal relationship between hyperparathyroidism and muscle wasting is not yet established. Interestingly, the activation of PTH–PTH1R signaling in adipocytes has been associated with muscle wasting or cachexia observed in conditions such as CKD and cancer [32,141]. In a scenario of persistently high parathyroid hormone (PTH) levels, PTH promotes the expression of UCP1, a thermogenic gene, in mice with CKD [32]. This leads to browning of white adipose tissue and energy generation. The browning process is associated with muscle atrophy and involves protein kinase A activation via PTH/PTHrP signaling, as well as muscle mass wasting through UPS. The interaction between muscle and fat plays a significant role in tissue catabolism induced by PTH or PTHrP [29]. Consequently, the development of muscle wasting under CKD conditions is significantly influenced by the signaling of the PTH and PTH receptor in adipose tissue.

## 5. Crosslinks between Bones and Muscles

Bone health relies on the coordinated actions of osteocytes, osteoblasts, and osteoclasts, which secrete bone-derived factors known as osteokines. Aging and metabolic disease can disrupt this process, leading to bone loss and an elevated risk of fractures [142]. Endocrine factors, genetics, and development have been extensively studied in the relationship between osteoporosis and sarcopenia. In particular, the role of myokines and osteokines has gained attention in understanding the pathogenesis of osteopenia.

Our recent clinical cohort found that ESRD patients had low handgrip strength, possibly due to higher levels of IS [143]. Elevated IS levels are associated with muscle issues, and incorporating AST-120 in CKD treatment shows promise for improving sarcopenia [144]. Another study revealed a positive correlation between IS levels and bone formation rate in predialysis CKD patients [106].

IS levels increase as renal function declines, with ESRD patients having significantly higher levels. According to a previous report, the IS concentrations are as follows: 0.97 μM at stage 1, 1.98 μM at stage 2, 12.73 μM at stage 3, 21.48 μM at stage 4, 78.79 μM (19.8 μg/mL) at stage 5, and 169.12 μM (44.86 μg/mL) at stage 5D [145]. In our previous study, we investigated the impact of IS on bone cells and their development. Using osteoclast precursor cells and Raw 264.7 cells, we treated osteoclast precursor cells with 100 μM of IS, which represents the average concentration found in the serum of patients with end-stage renal disease (ESRD, stage 5D). Additionally, we cultured Raw 264.7 cells with 50 ng/mL of soluble RANKL in various IS concentrations (0, 20, 100, 250, 500, and 1000 µM) to assess osteoclastogenesis at different IS levels. The results revealed that as the concentration of IS increased beyond 100 µM, there was a dose-dependent decrease in the percentage of TRAP-positive cells and the number of mature osteoclast cells. Furthermore, when the cells were exposed to IS for longer durations (5 days), the suppression of osteoclastogenesis became more pronounced. These findings indicate that the impact of IS on osteoclastogenesis is both concentration dependent and influenced by the duration of exposure [146].

In another study, we investigated the impact of IS on osteoblast development. Primary osteoblast cells were cultured in an osteogenesis medium and exposed to various concentrations of IS (0, 20, 100, 250, 500, and 1000 μM). The first set of cultures lasted for 14 days, during which we observed the effects of IS on osteoblast development using alkaline phosphatase staining, which reflects osteoblast activity. As the concentration of IS increased, we noticed a dose-dependent decrease in alkaline phosphatase activity, indicating impaired osteoblast development in response to higher IS concentrations. The second set of cultures continued for 21 days, and we examined the effects of IS on osteoblast mineralization using alizarin red staining. Once again, we found a dose-dependent decrease in alizarin red staining as the IS concentration increased. This finding suggests that the mineralization capacity of osteoblasts was adversely affected by higher levels of IS. Overall, this study demonstrated that increasing concentrations of IS negatively impact osteoblast development and mineralization, highlighting the potential detrimental effects of IS on bone health [147]. However, further investigation is necessary to fully understand the relationship between mineral and bone disorder and its impact on the balance of bone and muscle in CKD patients.

### 5.1. Osteokines and Muscle Atrophy

Fractures often occur before muscle and strength loss in elderly individuals with CKD. Similarly, extended bed rest or microgravity conditions can lead to rapid muscle mass loss along with bone loss [148]. Bone cells play a crucial role in detecting and converting mechanical signals into biochemical signals to regulate bone remodeling [149]. Osteocytes, with their dendritic protrusions, cellular bodies, and cilia, function as mechanosensory cells [150]. Mechanical stress triggers intracellular calcium release, ATP, NO, and prostaglandin release, affecting osteoblasts and osteoclasts [151]. Nitrous oxide inhibits osteoclasts, promoting bone mass increase, while ATP, nitrous oxide, and prostaglandin stimulate bone synthesis. Osteocytes activate the Wnt/β-catenin pathway in response to mechanical stress, enhancing osteogenesis [152]. The Wnt/β-catenin pathway is activated in response to mechanical stress, promoting osteogenesis. Osteoclasts regulate bone resorption through the RANKL/OPG system, while myostatin, follistatin, and irisin play roles in bone remodeling and muscle–bone crosstalk [153]. Irisin and leptin are believed to prevent bone loss by reducing osteoclast production.

#### Osteocalcin (OCN)

OCN, a marker of osteoblast bone formation, is a γ-carboxyglutamate protein secreted only by osteoblasts [154,155,156]. In primary osteoblast cultures, IS significantly inhibits mineralized bone formation by the inhibition of OCN and bone morphogenetic protein 2 (BMP2) mRNA expression [96]. OCN is thought to play an important role in glucose and energy homeostasis, increasing skeletal muscle degradation and nutrient absorption, and increasing exercise capacity [157,158]. OCN increases the expression of fatty acid transporters, thereby enhancing β-oxidation processes and translocation of glucose transporter 4 (GLUT4) to the plasma membrane while inducing glucose uptake and skeletal muscle degradation [159]. Otherwise, OCN induces muscle synthesis of IL-6 through a G protein-coupled receptor class C group 6 member A (GPRC6A) but also helps adaptation to exercise by producing OCN in bones [159,160].

### 5.2. Myokines Affect Bones

Multiple signaling factors from muscle and bone, including myostatin, irisin, β-aminoisobutyric acid (BAIBA), IL-6, IGF family, FGF-23, Wnt1, Wnt3a, PGE2, FGF9, RANKL, osteocalcin, and sclerostin, have been identified as potential mediator IAAs in biochemical and molecular interactions. Understanding these molecules and their mechanisms provides an exciting opportunity to develop new pharmaceutical approaches that can simultaneously target diseases occurring together, such as osteoporosis and sarcopenia [161].

## 6. Possible Therapeutic Considerations for Bone and Muscular Health in CKD

PBUTs can negatively affect skeletal and muscle functions, leading clinicians to recognize the effectiveness of antiuremic toxin medications. The subsequent text will explore various therapeutic considerations.

### Removal or Decrease of PBUT Precursor from the Intestinal Tract

AST-120, an oral spherical carbon adsorbent, absorbs protein-bound uremic toxin precursors from the intestine [162]. In uremic rats, AST-120 shows potential in preventing the decline of low bone turnover [101]. Previous studies have indicated that AST-120, when administered orally along with a low-protein diet, may slow the progression of renal function deterioration [69]. Advanced kidney disease patients often exhibit high levels of IS in the serum [163], but treatment with AST-120 can reduce serum IS levels compared with the control group with no treatment. Exercise capacity in skeletal muscle is positively correlated with mitochondrial function, which is mainly controlled by mitochondrial biosynthesis and degradation [164,165]. AST-120 treatment reduces IS levels, restoring mitochondrial function and improving exercise capacity in skeletal muscles [166].

Resveratrol (RSV) is a small polyphenol found in grape skins, cocoa, blueberries, and peanuts [167]. It improves the tight junction of the intestinal epithelium, providing benefits [168]. RSV inhibits hepatic sulfotransferase, reducing the synthesis of IS and offering cardiovascular protection [169]. In lower concentrations, RSV reduces reactive oxygen and nitrogen species formation in primary granulosa cells and promotes mitochondrial biogenesis [170,171]. RSV therapy promotes osteogenic differentiation and mitochondrial activity in bone marrow–derived mesenchymal stem cells [172]. Additionally, RSV influences the mitogen-activated protein kinase (MAPK) pathway, contributing to skeletal development [173].

Probiotics, living microorganisms, offer multiple benefits, such as restoring balance to the intestinal microbiota, improving mucosal integrity and cellular tight junctions, and regulating the microbiota environment by lowering pH. Indole and p-cresol, produced by the intestinal microbiota, bind to plasma albumin and are typically excreted in urine [174]. The accumulation of PBUTs in CKD disrupts gut microbiota and harms bone metabolism and cardiovascular health [174,175]. PBUTs generate excessive reactive oxygen species that impair osteoblast and osteoclast function and induce PTH resistance in bone cells [175]. Probiotics can mitigate IS toxicity, protect cardiovascular function, and decelerate renal deterioration.

Probenecid, an inhibitor of the organic anion transporter (OAT), demonstrated a dose-dependent blocking effect on IS inhibition in a previous mouse study [93]. IS enters cells via OAT-3 and promotes oxidative stress, leading to alterations in osteoblast function and decreased PTHR expression [7]. Probenecid protects osteoblasts from IS-induced damage and stabilizes osteoclasts, promoting favorable bone metabolism.

## 7. Other Possible Therapeutic Strategies

Metabolic acidosis, aluminum toxicity, and diabetes are factors associated with bone and muscle loss. It is important to treat hyperglycemia, avoid aluminum-containing phosphate binders, and correct metabolic acidosis, especially in CKD patients. Ursolic acid has shown promise in improving mitochondrial biosynthesis and mitigating muscle dysfunction caused by CKD by influencing mouse myoblast differentiation, IL-6 secretion, and ATP levels [176]. Furthermore, the reduction of hyperphosphatemia has been associated with decreased inflammation and improvement in anemia and skeletal muscle wasting [177].

A previous study showed that combining omega-3 fatty acid and menaquinone-7 supplementation in uremic rats can help prevent aortic vascular calcification, reduce osteoclast activation in the bone, and improve sarcopenia-related molecules [178]. Nutrition plays a significant role in sarcopenia. A clinical study found that the elderly hemodialysis sarcopenia index (EHSI) can identify sarcopenia diagnosed by the European Working Group on Sarcopenia in Older People (EWGSOP) second meeting using easily accessible anthropometric and nutritional parameters. A combination of gender, age, serum albumin, phosphate, and cholesterol can effectively predict the severity of sarcopenia [179]. The current nutritional therapy for uremic sarcopenia includes oral supplements, parenteral nutrition, enteral nutrition, high protein and fiber diet, and gastrectomy [180].

Nutritional interventions, along with exercise, have been shown to have positive effects on measures of sarcopenia in individuals with CKD. Longer intervention duration, supervision, and participant adherence increase muscle strength outcomes. CKD patients may require higher intensity and progressive loading in resistance exercise to see noticeable results in muscle mass. Although the current evidence for progressive resistance exercise in CKD is encouraging, real-life applications in clinical settings are limited. A multidisciplinary patient-centered approach with regular follow-up may be beneficial for managing the complexity of sarcopenia in CKD [181].

Combining targeted exercise with personalized dietary–nutritional therapy is the ideal approach for treating uremic sarcopenia. The key interventions for preventing and treating sarcopenia in CKD include aerobic and resistance exercises, along with personalized dietary–nutritional interventions. However, the effectiveness of these interventions in treating sarcopenia and preventing clinical consequences in CKD patients is still being determined [180,182].

## 8. Conclusions

Bone–muscle crosstalk, mediated by factors in the bloodstream, influences the physiology of various organs. The term “osteosarcopenia” refers to the combination of low bone density (osteopenia) and reduced muscle mass. In CKD, different stages have distinct impacts on bone and muscle homeostasis. In early CKD, the accumulation of PBUTs significantly inhibits bone and muscle function. As CKD progresses to the late stage, increased PBUT levels further impede bone turnover and worsen muscle wasting and atrophy. Meanwhile, hyperparathyroidism exacerbates PBUT-induced muscle damage but can improve low bone turnover. We propose that, in CKD, targeting PBUTs in both the early and late stages can have beneficial effects on bone and muscle. In the late stage of CKD, targeting high PTH levels can prevent bone loss associated with high bone turnover and mitigate muscle wasting caused by prolonged exposure to high levels of PTH. Although PBUTs have been implicated in oxidative stress, inflammation, and fibrosis in cardiovascular and renal tissues, their exact role and contribution to CKD-related problems require further investigation. The metabolism and elimination of PBUTs along the diet–gut–liver–kidney axis are critical areas of study. While PBUTs are important in CKD, additional research is necessary to fully understand their impact on all problems faced by CKD patients.

## Figures and Tables

**Figure 1 biomedicines-11-02076-f001:**
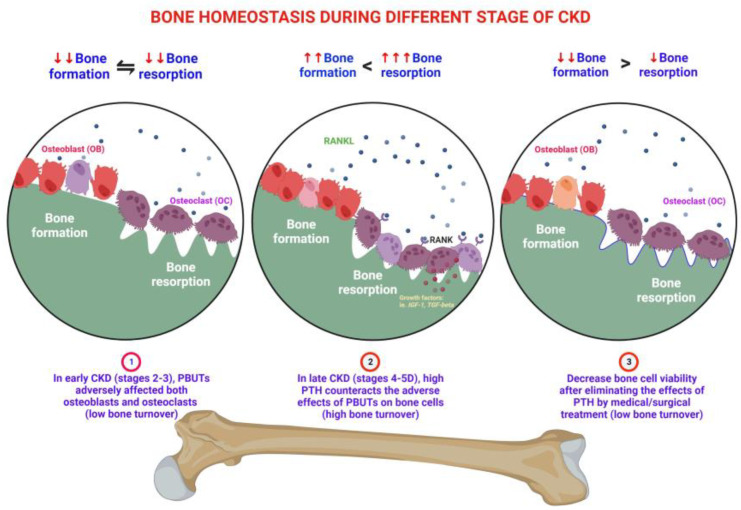
PBUTs and PTH affect bone metabolism at different stages of CKD. ① In the early stages of CKD (stages 2–3), the release of Wnt signaling inhibitors (DKK1, SOST, and sFRP) from the kidney or bone cells reduces the viability of osteoblasts. Additionally, PBUTs, such as IS and pCS, impair the function of osteoblasts and osteoclasts. CKD affects the gut microbiome, leading to the accumulation of uremic toxins in the blood, particularly IS, which is associated with low bone turnover, suppressed bone formation, and apoptosis in osteoblasts. This condition is known as “uremic osteoporosis”. Further bone loss occurs due to the progressive deterioration of renal function, metabolic acidosis, and hyponatremia. ② In CKD stages 4–5, the dysregulation of calcium, phosphate, vitamin D, and PTH can result in varying levels of PTH. High-turnover bone disease develops when elevated serum PTH levels override bone formation inhibitors, leading to increased osteoclast and osteoblast activity. ③ However, treating SHPT can restore bone cells to their original low viability status and low bone turnover status.

**Figure 2 biomedicines-11-02076-f002:**
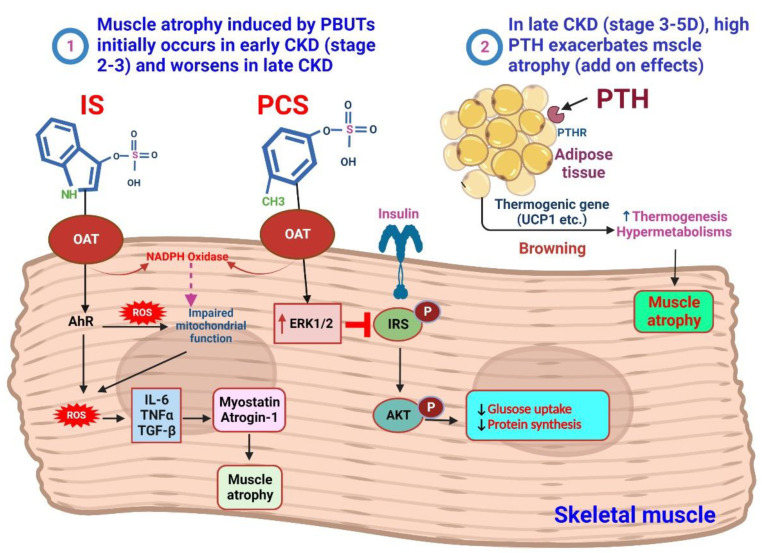
Mechanism of PBUTs and PTH-mediated skeletal muscle atrophy. Redox signaling changes in CKD, caused by uremic toxins, inflammation, and metabolic/hormonal shifts, resulting in oxidative stress, leading to muscle wasting and bone loss. ① Mild accumulation of PBUTs significantly contributes to muscle wasting in early-stage CKD (stages 2–3). Molecular mechanisms involved include imbalanced protein degradation and synthesis, increased reactive ROS and inflammatory cytokines, activation of myostatin and atrogene expression, impaired mitochondrial function, and negative effects from uremic toxins, parathyroid hormone, glucocorticoids, and angiotensin II. IS enters muscle cells using an OAT, where IS stimulates the pathway of AhR and NADPH oxidase to induce the production of ROS. Excessive production of ROS will produce inflammatory cytokines when myostatin and atrogin-1 are overexpressed, and this will be linked to muscle atrophy. The production of ROS has an impact on mitochondrial function. pCS induces insulin resistance in muscles due to decreased insulin or IGF-1-IRS-AKT capacity triggered by ERK1/2 activity. In late CKD (stages 4–5D), cachexia is a syndrome characterized by further increased energy expenditure and loss of muscle and adipose tissues. In addition to the further increase in PBUT levels, the excessive levels of PTH have detrimental effects on skeletal muscle metabolism, leading to aggravated muscle weakness and atrophy. ② High PTH levels are associated with sarcopenia and contribute to muscle loss. PTH and PTHrP drive adipose tissue browning and muscle wasting in cachexia. Elevated PTH indirectly leads to decreased muscle protein synthesis by acting on PTH receptors expressed in adipose tissue, thereby activating the expression of thermogenic genes, and finally causing muscle hypermetabolism and atrophy.

## Data Availability

This is a narrative review article. The primary collection of documents for analysis and review comes from PubMed.

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
