# Peer review of "The Potential Influence of Uremic Toxins on the Homeostasis of Bones and Muscles in Chronic Kidney Disease"

_biomedicines, 2023, doi:10.3390/biomedicines11072076_

Round 1

Reviewer 1 Report

The authors have done a comprehensive analysis of all the available knowledge on molecular mechanisms of CKD and the role of PBUTs in the disease.

Comments

1.      The title does not correspond to the presented text as sections 6.2-6.5; 5.2.1-5.2.5; 5.1.2-5.1.6;5.1; 4.3; 4.2.2-4.2.5; 4.2 contain no information of PBUTs. The authors should either change the title or remove data not related to PBUTs.

2.      Section 2: The authors should present any data on the causes of PBUTs accumulation in CKD patients.

3.      All the abbreviations should be disclosed on first use. This should be corrected.

4.      Section 2.2.1, etc: The schematic for all signaling pathway descriptions should be presented.

5.      5. Lines 121-122; 129-130; 137-139; 298-299; 401-402; 755-756: These sentences are not clear. They should be rephrased.

6.      Line132 and 134: The term toughness should be disclosed.

7.      Line 256: The correct writing of “collagen-1” is type I collagen. This should be corrected.

8.      Sections 2.2.3; 5.2.5 should be rephrased as they contain repeats and inconsistent statements.

9.      Lines 221; 242; 253; 335 References are required at the end of these sentences.

10.  Section 4.2 has no references. This should be corrected.

11.  Figure 1 is not related to the subject of the review. This should be corrected.

12.  Line 761;767;768: Conclusion should not contain any references. It should summarize all the data presented before.

13.  Line 762: The authors should prove that all the problems of CKD patients result for the presence of PBUTs.

1.       Lines 121-122; 129-130; 137-139; 298-299; 401-402; 755-756: These sentences are not clear. They should be rephrased.

2.      Line132 and 134: The term toughness should be disclosed.

3.      Line 256: The correct writing of “collagen-1” is type I collagen. This should be corrected.

4.      Sections 2.2.3; 5.2.5 should be rephrased as they contain repeats and inconsistent statements.

Author Response

Response to Reviewer 1

Reviewer 1

Comments and Suggestions for Authors

The authors have done a comprehensive analysis of all the available knowledge on molecular mechanisms of CKD and the role of PBUTs in the disease.

Comments:

  1. The title does not correspond to the presented text as sections 6.2-6.5; 5.2.1-5.2.5; 5.1.2-5.1.6;5.1; 4.3; 4.2.2-4.2.5; 4.2 contain no information of PBUTs. The authors should either change the title or remove data not related to PBUTs.

Response: We appreciate the valuable suggestions provided by the reviewers. We have carefully considered their criticisms and learned a great deal from them. To ensure the credibility of the text, we have removed sections that do not correspond to the contents. We have retained Section 4.3 as requested. In late-stage CKD (stage 3-5D), an increase in PTH levels can help alleviate the inherent low bone turnover. However, it worsens the muscle wasting caused by accumulated PBUTs. Therefore, in this context, increased PTH levels have a contrasting effect on bone turnover and muscle wasting.

  1. Section 2: The authors should present any data on the causes of PBUTs accumulation in CKD patients.

Response: Thank you for your critical comments. We have included a statement in the description regarding the causes of protein-bound uremic toxins (PBUTs) accumulation in chronic kidney disease (CKD) patients.

Protein-bound uremic toxins (PBUTs) accumulate in chronic kidney disease (CKD) patients due to the impaired renal clearance of these toxins. PBUTs are a group of solutes that are normally excreted by healthy kidneys but accumulate in CKD due to decreased filtration and impaired tubular secretion, gut microbial metabolism, and dietary precursors [PMID: 35050985]. PBUTs such as indoxyl sulfate (IS) and p-cresyl sulfate (pCS), tightly bind to proteins like serum albumin, making their removal from the body challenging [PMID: 34200937].

  1. All the abbreviations should be disclosed on first use. This should be corrected.

Response: We appreciate the valuable comments provided by the reviewer. In the revised manuscript, all abbreviations have been disclosed on their first use.

  1. Section 2.2.1, etc: The schematic for all signaling pathway descriptions should be presented.

Response: We value the reviewer's critical feedback and have incorporated the necessary revisions suggested in their comments. Additionally, we have provided the essential schemata required for the manuscript.

  1. 5. Lines 121-122; 129-130; 137-139; 298-299; 401-402; 755-756: These sentences are not clear. They should be rephrased.

Response: To ensure better comprehension by the readers, we have rephrased all the unclear sentences as mentioned by the reviewer. We express our gratitude for the valuable feedback provided by the reviewer.

  1. Line132 and 134: The term toughness should be disclosed.

Response: To ensure better comprehension by the readers, we have changed the term "toughness" to "strength" in our revised manuscript. We express our gratitude for the valuable feedback provided by the reviewer on our manuscript.

  1. Line 256: The correct writing of “collagen-1” is type I collagen. This should be corrected.

Response: The mistake has been rectified.

  1. Sections 2.2.3; 5.2.5 should be rephrased as they contain repeats and inconsistent statements.

Response: To ensure better comprehension by the readers, we have rephrased all the unclear sentences as mentioned by the reviewer. We express our gratitude for the valuable feedback provided by the reviewer.

  1. Lines 221; 242; 253; 335 References are required at the end of these sentences.

Response: The references have been added.

  1. Section 4.2 has no references. This should be corrected.

Response: After rewriting the paragraph, the references were included.

  1. Figure 1 is not related to the subject of the review. This should be corrected.

Response: Based on the reviewer's opinion, Figure 1 was replaced with a new one.

  1. Line 761;767;768: Conclusion should not contain any references. It should summarize all the data presented before.

Response: In response to the reviewer's feedback, we have removed all the references from the conclusion section.

  1. Line 762: The authors should prove that all the problems of CKD patients result for the presence of PBUTs.

Response: We sincerely appreciate the valuable comment provided by the reviewer. In response, we have included additional statements to elucidate the relationship between the presence of protein-bound uremic toxins (PBUTs) and the problems experienced by patients with CKD.

To prove that all the problems of CKD (chronic kidney disease) patients result from the presence of protein-bound uremic toxins (PBUTs), comprehensive research and evidence are necessary. While the provided information does not directly address the specific question, it offers insights into the accumulation of PBUTs in CKD patients and their impact on health. Several studies highlight the importance of PBUTs in CKD and their association with various biological problems. Although PBUTs have been implicated in oxidative stress, inflammation, and fibrosis processes in both cardiovascular and renal tissues, their exact role in uremic pathophysiology and the extent of their contribution to all problems faced by CKD patients require further investigation. Additionally, the metabolism and elimination of PBUTs along the diet-gut-liver-kidney axis have been highlighted as crucial areas for study. In summary, while the provided information indicates the importance of PBUTs in CKD and their association with health problems, further research is necessary to definitively prove that all the problems of CKD patients result from the presence of PBUTs.

Comments on the Quality of English Language

  1. Lines 121-122; 129-130; 137-139; 298-299; 401-402; 755-756: These sentences are not clear. They should be rephrased.
  2. Line132 and 134: The term toughness should be disclosed.
  3. Line 256: The correct writing of “collagen-1” is type I collagen. This should be corrected.
  4. Sections 2.2.3; 5.2.5 should be rephrased as they contain repeats and inconsistent statements.

Response: We have addressed and resolved all the above mentioned questions concerning the Quality of English Language.

Reviewer 2 Report

Patients with chronic kidney disease (CDK) and increased protein-bound uremic toxins (PBUTs) may suffer from osteosarcopenia, the combination of osteopenia and low muscle mass. The authors assume that interactions between muscles and bones through the release of different factors in individual tissues may play an important role in the mutual modulation in CKD. To explain these interferences the authors describe a great number of possible interactions of PBUTs with cellular functions such as the connection between AhR, the immune system, and bone remodeling in CKD patients or that AhR signaling plays a role in diabetic nephropathy, causing oxidative stress injury, mesangial cell activation, extracellular matrix accumulation, and macrophage infiltration due to AhR nuclear translocation and increased toxic bioproducts and many other possible effects.

Although these interactions seem plausible no information is provided under what conditions each of them may become effective. For example, oxidative stress being repeatedly mentioned, only starts, when the antioxidative mechanisms become overwhelmed.

In summary, although potential mechanisms resulting from increased PBUTs are described, no information is provided when and under what condition a specific cascade is triggered. Without this the manuscript presents a bulk of information without coherence, which is of no added value and does not warrant publication.

A way out may be that the authors focus on the specific effects of PBUTs on the homeostasis of bones and muscles. 

Author Response

Response to Reviewer 2

Comments and Suggestions for Authors

Patients with chronic kidney disease (CDK) and increased protein-bound uremic toxins (PBUTs) may suffer from osteosarcopenia, the combination of osteopenia and low muscle mass. The authors assume that interactions between muscles and bones through the release of different factors in individual tissues may play an important role in the mutual modulation in CKD. To explain these interferences the authors describe a great number of possible interactions of PBUTs with cellular functions such as the connection between AhR, the immune system, and bone remodeling in CKD patients or that AhR signaling plays a role in diabetic nephropathy, causing oxidative stress injury, mesangial cell activation, extracellular matrix accumulation, and macrophage infiltration due to AhR nuclear translocation and increased toxic bioproducts and many other possible effects.

Although these interactions seem plausible no information is provided under what conditions each of them may become effective. For example, oxidative stress being repeatedly mentioned, only starts, when the antioxidative mechanisms become overwhelmed.

In summary, although potential mechanisms resulting from increased PBUTs are described, no information is provided when and under what condition a specific cascade is triggered. Without this the manuscript presents a bulk of information without coherence, which is of no added value and does not warrant publication.

A way out may be that the authors focus on the specific effects of PBUTs on the homeostasis of bones and muscles.

Response: We appreciate the valuable comments provided by the reviewer. We agree with the reviewer's suggestion regarding the need to provide information about the specific cascade triggered by increased PBUTs and the corresponding conditions. In response to this feedback, we will address this drawback by including a necessary statement in the revised "Introduction" section as follows:

In early chronic kidney disease (CKD stages 2-3), Wnt signaling inhibitors (DKK1, SOST, and sFRP) are released from the kidney or bone cells. These inhibitors can reduce the viability of osteoblasts. Protein-bound uremic toxins (PBUTs) like IS/PCS also impair the function of osteoblasts and osteoclasts. CKD affects the gut microbiome, leading to the accumulation of uremic toxins in the blood. Indoxyl sulfate (IS) is linked to low bone turnover, suppressing bone formation, and promoting apoptosis in osteoblasts. "Uremic osteoporosis" refers to bone quality loss despite normal bone mass due to uremic toxins. Progressive deterioration of renal function, metabolic acidosis, and hyponatremia contribute to further bone loss. In CKD stages 4-5, dysregulation of calcium, phosphate, vitamin D, and PTH can result in varying levels of PTH. High-turnover bone disease occurs when increased serum PTH levels overcome bone formation inhibitors. High PTH levels increase osteoclasts and osteoblasts. However, treating secondary hyperparathyroidism (SHPT) can restore bone cells to their original low viability status and low bone turnover disorders.

Redox signaling changes in CKD, caused by uremic toxins, inflammation, and metabolic/hormonal shifts, resulting in oxidative stress, leading to muscle wasting and bone loss. Mild accumulation of protein-bound uremic toxins (PBUTs) significantly contributes to muscle wasting in early-stage CKD (stage 2-3). Exercise can potentially mitigate these effects. Although antioxidant therapies have shown promise in experimental CKD models, treating CKD-related muscle wasting and bone loss remains a challenge. CKD is also linked to muscle wasting and reduced endurance. Molecular mechanisms involved include imbalanced protein degradation and synthesis, increased reactive oxygen species (ROS) and inflammatory cytokines, activation of myostatin and atrogenes expression, impaired mitochondrial function, and negative effects from uremic toxins, parathyroid hormone, glucocorticoids, and angiotensin II. Indoxyl sulfate is specifically implicated in stimulating ROS-mediated myostatin and atrogenes expression and impairing mitochondrial function. Our recent cohort study in ESRD patients found that low handgrip strength was associated with higher indoxyl sulfate and lower irisin concentrations compared to the general population. In late CKD (stage 4-5D), cachexia is a syndrome characterized by further increased energy expenditure and loss of muscle and adipose tissues. In addition to the further increase of PBUT levels, the excessive levels of parathyroid hormone (PTH) have detrimental effects on skeletal muscle metabolism, leading to aggravated muscle weakness and atrophy. High PTH levels are associated with sarcopenia in elderly women and contribute to muscle loss in aging. PTH and PTH-related peptides (PTHrP) drive adipose tissue browning and muscle wasting in cachexia. Targeting the PTH/PTHrP pathway could be a potential therapeutic approach for treating cachexia observed in chronic kidney disease (CKD) and cancer. We have changed Figure 1 to explain the above statements.

Based on the varying effects of PBUTs and PTH levels on bone turnover and muscle wasting in CKD, we conducted this review to explore the specific impact of PBUTs on the homeostasis of bones and muscles in CKD.

In addition, Based on the reviewer’s comments, we have changed our manuscript title to "Impact of Uremic Toxins on the Homeostasis of Bones and Muscles in Chronic Kidney Disease".

Reviewer 3 Report

As always, a review of the information related to the topic of interest (sarcopenia, bone loss and uremia), is a task that deserves recognition.

The authors collect information about patients with chronic kidney disease (CKD) who have often have elevated levels of protein-bound uremic toxins (PBUTs) that cause alterations as inflammation, oxidative stress, muscle mass and bone density reductions, in addition to worsening kidney function.

The text describes the pathophysiology of these toxins as well as the alterations they cause in various cell signaling processes. It also describes issues related to bone and muscle metabolism and their interactions in order to understand the mechanisms between bone and muscle loss in CKD and suggest new therapies for musculoskeletal diseases, specifically for bone loss and muscle atrophy.

The structure of the manuscript and the collection of information seem adequate to me. Perhaps the incorporation of some images or diagrams will make the reading more enjoyable and the information will be more illustrative.

The authors organized the information as follows:

1. Introduction

2. Pathophysiology of PBUTs: indoxyl sulfate (IS) and p-cresyl sulfate (PCS)

2.1. Metabolism of PBUTs

2.2. Pathogenesis of PBUTs

2.2.1. Aryl hydrocarbon receptor (AhR) signalling

2.2.2. PBUTs enhance reactive oxygen species (ROS)

2.2.3. PBUTs diminish the synthesis of nitric oxide (NO)

2.2.4. The epigenetic effects of PBUTs

3. Effects of PBUTs on bone

3.1. PBUTs influence bone metabolism

3.1.1. Uremic toxin exposure affects osteoclastogenesis

3.1.2. PBUTs impair osteoblastogenesis

3.1.3. PBUTs reduce bone mass

3.1.4. PBUTs reduce the bone quality

3.1.5. PBUTs induce bone resistance to parathyroid hormone (PTH)

3.1.6. PBUTs disturb the synthesis of vitamin D

3.1.7. PBUTs affect the differentiation of T cells

4. IS and PCS on the muscle

4.1. Sarcopenia in chronic kidney disease

4.2. Mechanisms of muscle wasting in CKD

4.2.1. Metabolic acidosis

4.2.2. PBUTs disrupt insulin/Insulin-like growth factor-1 (IGF-1) signaling

4.2.3. Inflammation and sarcopenia

4.2.4. microRNAs induce sarcopenia

4.2.5. Advanced glycosylation end-products (AGEs)

4.3. Parathyroid hormone and muscle atrophy

5. Crosslinks between bones and muscles

5.1. Osteokines and muscle atrophy (Figure 1)

5.1.1. Osteocalcin (OCN)

5.1.2. Transforming growth factor-β (TGF-β)

5.1.3. Sclerostin (SOST)

5.1.4. Receptor activator of nuclear factor Κ-B ligand (RANKL)

5.1.5. Wnt signaling pathway

5.1.6. Fibroblast growth factors (FGFs)

5.1.7. Prostaglandin E2 (PGE2)

5.2. Myokines affect bones

5.2.1. Myostatin (MSTN)

5.2.2. Insulin-like growth factor-1 (IGF-1)

5.2.3. Interleukins (ILs)

5.2.4. Irisin

5.2.5. β-aminoisobutyric acid (BAIBA)

6. Possible therapeutic considerations for bone and muscular health in CKD

6.1. Removal or decrease of PBUTs precursor from the intestinal tract

6.2. Vitamin D supplement

6.3. Anti-SOST antibody

6.4. ActRIIB-Fc

6.5. Reducing oxidative stress

7. Other possible therapeutic strategies

8. Conclusions

The issues covered range from the general to the specific and, as expected, therapeutic alternatives for this condition are proposed. However, I suggest that nutrient supplementation and exercise should be included as alternative therapies, as both have been shown to play important roles in this disease:

·      Lee SM, Jeong EG, Jeong YI, Rha SH, Kim SE, An WS. Omega-3 fatty acid and menaquinone-7 combination are helpful for aortic calcification prevention, reducing osteoclast area of bone and Fox0 expression of muscle in uremic rats. Ren Fail 2022;44(1):1873-1885.

·      Noce A, Marrone G, Ottaviani E, et al. uremic sarcopenia and its possible nutritional approach. Nutrients 2021;13(1):147.

·      Sánchez-Tocino ML, Mas-Fontao S, Gracia-Iguacel C, Pereira M, González-Ibarguren I, Ortiz A, Arenas MD, Parra EG. A sarcopenia index derived from malnutrition parameters in elderly haemodialysis patients. Nutrients 2023; 15(5):1115.

·      Chatzipetrou V, Bégin MJ, Hars M et al. Sarcopenia in chronic kidney disease: A scoping review of prevalence, risk factors, association with outcomes, and treatment. Calcif Tissue Int 2022;110:1-31.

·      Noor H, Reid J, Slee A. Resistance exercise and nutritional interventions for augmenting sarcopenia outcomes in chronic kidney disease: a narrative review. J Cachexia Sarcopenia Muscle 2021;12(6):1621-1640.

The two hundred and seventy references cited range from the year 1996 to 2023, which is curious considering that nineteen are from the year 1996 to the year 2000. Perhaps it is appropriate to indicate that some references are quite old.

Among the vast number of reports on the subject addressed about sarcopenia, bone, and uremia, it is logical that there is a lack of references that could be useful. Examples of them could be the following:

·      Kim DW, Song SH. Sarcopenia in chronic kidney disease: from bench to bedside. Korean J Intern Med 2023;38(3):303-321.

·      Mohanasundaram S, Fernando E. Uremic Sarcopenia. Indian J Nephrol 2022;32(5):399-405.

·      Nishi H, Takemura K, Higashihara T, Inagi R. Uremic Sarcopenia: Clinical Evidence and Basic Experimental Approach. Nutrients 2020;12(6):1814.

Author Response

Response to reviewer 3

Comments and Suggestions for Authors

As always, a review of the information related to the topic of interest (sarcopenia, bone loss and uremia), is a task that deserves recognition.

The authors collect information about patients with chronic kidney disease (CKD) who have often have elevated levels of protein-bound uremic toxins (PBUTs) that cause alterations as inflammation, oxidative stress, muscle mass and bone density reductions, in addition to worsening kidney function.

The text describes the pathophysiology of these toxins as well as the alterations they cause in various cell signaling processes. It also describes issues related to bone and muscle metabolism and their interactions in order to understand the mechanisms between bone and muscle loss in CKD and suggest new therapies for musculoskeletal diseases, specifically for bone loss and muscle atrophy.

The structure of the manuscript and the collection of information seem adequate to me. Perhaps the incorporation of some images or diagrams will make the reading more enjoyable and the information will be more illustrative.

  1. Response: We have included two figures and corresponding legends to elucidate the impact of PBUTs on bone and muscle at different stages of CKD in our revised manuscript.

The issues covered range from the general to the specific and, as expected, therapeutic alternatives for this condition are proposed. However, I suggest that nutrient supplementation and exercise should be included as alternative therapies, as both have been shown to play important roles in this disease:

Lee SM, Jeong EG, Jeong YI, Rha SH, Kim SE, An WS. Omega-3 fatty acid and menaquinone-7 combination are helpful for aortic calcification prevention, reducing osteoclast area of bone and Fox0 expression of muscle in uremic rats. Ren Fail 2022;44(1):1873-1885.

Noce A, Marrone G, Ottaviani E, et al. uremic sarcopenia and its possible nutritional approach. Nutrients 2021;13(1):147.

Sánchez-Tocino ML, Mas-Fontao S, Gracia-Iguacel C, Pereira M, González-Ibarguren I, Ortiz A, Arenas MD, Parra EG. A sarcopenia index derived from malnutrition parameters in elderly haemodialysis patients. Nutrients 2023; 15(5):1115.

Chatzipetrou V, Bégin MJ, Hars M et al. Sarcopenia in chronic kidney disease: A scoping review of prevalence, risk factors, association with outcomes, and treatment. Calcif Tissue Int 2022;110:1-31.

Noor H, Reid J, Slee A. Resistance exercise and nutritional interventions for augmenting sarcopenia outcomes in chronic kidney disease: a narrative review. J Cachexia Sarcopenia Muscle 2021;12(6):1621-1640.

  1. Response: Thank you for the reviewer's critical comments. We greatly appreciate the valuable feedback provided. In response to the reviewer's suggestions, we have included novel information in our revised manuscript. This addition enriches the content and enhances the overall quality of our manuscript. (Section 7)

The two hundred and seventy references cited range from the year 1996 to 2023, which is curious considering that nineteen are from the year 1996 to the year 2000. Perhaps it is appropriate to indicate that some references are quite old.

  1. Response: Thank you for the reviewer's comment. We greatly appreciate the valuable feedback provided. Based on the reviewer's invaluable suggestions, we have made a necessary change to the manuscript. Specifically, we have addressed the issue related to the old references by deleting them from the reference list. This action ensures that our references are up-to-date and relevant to the topic discussed.

Among the vast number of reports on the subject addressed about sarcopenia, bone, and uremia, it is logical that there is a lack of references that could be useful. Examples of them could be the following:

  Kim DW, Song SH. Sarcopenia in chronic kidney disease: from bench to bedside. Korean J Intern Med 2023;38(3):303-321.

  Mohanasundaram S, Fernando E. Uremic Sarcopenia. Indian J Nephrol 2022;32(5):399-405.

  Nishi H, Takemura K, Higashihara T, Inagi R. Uremic Sarcopenia: Clinical Evidence and Basic Experimental Approach. Nutrients 2020;12(6):1814.

  1. Response: T Thank you for the valuable feedback from the reviewer. We have carefully considered the reviewer's suggestions and made the necessary changes. We have also incorporated numerous new references based on the reviewer's recommendations. These changes and additions have significantly enhanced the quality and credibility of our work. We appreciate the reviewer's input and acknowledge the invaluable contribution they have made to our manuscript. (Section 4.1)

Round 2

Reviewer 2 Report

The authors have collected an abundant amount of information to understand the mechanisms which impair homeostasis of bones and muscles in chronic kidney disease. Most of the information stems from animal experiments or in vitro studies at high doses with questionable relevance to humans. For example, line 193/194 describes that TCDD and benzo[59] pyrene (BaP) (please correct !) smoke toxins activate AhR to accelerate osteoclast bone resorption by activating cytochrome P450 1a/1b enzymes. This study (ref. 60) is an intro/in vivo (mice) study on BP, whereas TCCD was studied in vitro only, both at relatively high doses/concentrations. Since I am not aware that even after the Seveso accident bone health has been affected (Eskenazi et al 2018, Envir Int 121, 71-84) the impact of TCCD on bones in humans is questionable. Thus, the problem of the manuscript is that effects found in vitro and in vivo are described without an evaluation whether they are relevant for human exposure.

My recommendation is that all studies presented are scrutinized for human relevance and the relevant ones only are used to evaluate the potential impact of uremic toxins on bone and muscle homeostasis. Although this is a challenging task it helps to understand the actual mechanisms relevant for humans. Otherwise, the manuscript remains an accumulation of data without a critical evaluation.

Abbreviations. List of abbreviations is recommended.

In general abbreviations need to be explained once and then used only. Not either full name or abbreviation throughout the text. Occasionally even both are used. This needs to be harmonized. For example:

Indoxyl sulfate (IS): Although abbrev. IS is given when first time mentioned, the full name is repeatedly used in the text.

Also, PBUTs repeatedly explained, again in Fig. 1.

Line 179: worsen inflammation? Better: increase inflammation.

See above

Round 3

Reviewer 2 Report

There is no question that in CKD indoxyl sulfate (IS) levels increase and there is an association between IS and decreased bone formation in these patients.

To evaluate the mechanisms IS has been applied to osteoblasts in vitro without considering whether the effective in vitro concentrations are ever reached in vivo in CKD patients. This (one or several high concentrations) usually applies to any other in vitro studies. This questions the relevance of the observations for humans unless based on a concentration-effect curve, identification of a NOEL and a discussion whether the effective concentrations can ever be reached in exposed humans is evaluated and demonstrated. Therefore, my request is that the in vitro studies used to understand the potential mechanisms should be scrutinized for there in vivo relevance. Otherwise the information is of no practical value.

The authors at least should demonstrate that the IS concentrations used in the in vitro experiments are comparable to those in the bones of CKD patients.

Author Response

Response:

We greatly appreciate the constructive criticism from the reviewer. The reviewer's comments have provided us with valuable insights, and we agree with their important points. We have taken into consideration the feedback and made the following revisions in the manuscript (Line 502-532):

IS levels increase as renal function declines, with ESRD patients having signifi-cantly higher levels. According to a previous report, the IS concentrations are as follows: 0.97 μM at stage 1, 1.98 μM at stage 2, 12.73 μM at stage 3, 21.48 μM at stage 4, 78.79 μM (19.8 μg/ml) at stage 5, and 169.12 μM (44.86 μg/ml) at stage 5D [ref 145, Supplemental Table 4].

In our previous study, we aimed to investigate the impact of indoxyl sulfate (IS) on bone cells and their development. To achieve this, we conducted a series of experiments using osteoclast precursor cells (cultured Raw 264.7 cells). Initially, we exposed osteoclast precursor cells to IS at a concentration of 100 μM, which corresponds to the average concentration found in the serum of patients with end-stage renal disease (ESRD, stage 5D). Subsequently, we cultured Raw 264.7 cells in different concentrations of IS (0, 20, 100, 250, 500, and 1000 µM) along with 50 ng/mL of soluble Receptor Activator of Nuclear Factor-κB Ligand (sRANKL). This allowed us to observe osteoclastogenesis at various IS levels.

The results of our study revealed interesting findings. As the concentration of IS increased beyond 100 µM, we observed a dose-dependent decrease in both the percentage of TRAP-positive cells (undifferentiated osteoclast cells) and the number of mature osteoclast cells. Additionally, when the cells were exposed to IS for a longer duration (5 days), the suppression of osteoclastogenesis became more pronounced. This indicates that the effect of IS on osteoclastogenesis is not only concentration-dependent but also influenced by the duration of exposure.

The underlying mechanisms of how IS impacts osteoclastogenesis were also investigated in our study. The Aryl hydrocarbon receptor (AhR) played a key role in this process since IS acts as an endogenous AhR agonist. Our data demonstrated that both osteoclastogenesis and NFATc1 (nuclear factor of activated T-cells, cytoplasmic 1) expression were affected by IS through AhR signaling in dose- and time-dependent manners. Specifically, osteoclast differentiation increased with short-term, low-dose IS exposure but decreased with long-term, high-dose IS exposure. Furthermore, different IS concentrations influenced the role of AhR, switching it from a ligand-activated transcription factor to an E3 ubiquitin ligase. The AhR nuclear translocator also played a critical role in regulating these dual functions of AhR under IS treatment.

These findings shed light on the intricate relationship between IS, AhR signaling, and osteoclastogenesis. Understanding the IS/AhR/NFATc1 signaling axis in osteoclastogenesis can have important implications for patients with chronic kidney disease (CKD) [ref 146]. The alterations in bone turnover due to high IS levels in CKD patients have been linked to skeletal resistance to parathyroid hormone, oxidative stress in osteoblasts, and decreased bone turnover rate. Our study contributes to the potential role of AhR in the pathology and abnormality of bone turnover in CKD patients, providing valuable insights into the management of bone-related complications in this population.

In another study, we investigated the impact of indoxyl sulfate (IS) on osteoblast development. To conduct this research, we cultured primary osteoblast cells in an osteogenesis medium and exposed them to various concentrations of IS, including 0, 20, 100, 250, 500, and 1000 μM. Our study was divided into two sets of cultures, each with a different focus. The first set of cultures lasted for 14 days, during which we observed the effects of IS on osteoblast development using alkaline phosphatase staining, which is an indicator of osteoblast activity. As the concentration of IS increased, we noticed a dose-dependent decrease in alkaline phosphatase activity. This finding suggests that higher concentrations of IS led to impaired osteoblast development. The second set of cultures continued for 21 days, during which we examined the effects of IS on osteoblast mineralization (function) using Alizarin Red staining. Once again, we found a dose-dependent decrease in Alizarin Red staining as the concentration of IS increased. This outcome indicates that higher levels of IS adversely affected the mineralization capacity of osteoblasts [ref 147].

Our findings demonstrate that increasing concentrations of indoxyl sulfate have a negative impact on both osteoblast development and mineralization. These results highlight the potential detrimental effects of IS on bone health and emphasize the importance of understanding and managing the toxic effects of uremic toxins like IS on the skeletal system. As IS is associated with chronic kidney disease and its complications, including bone mineral disease, further research in this area could provide valuable insights for preventing and treating bone-related complications in CKD patients.
